# Profile of Small RNAs, vDNA Forms and Viral Integrations in Late Chikungunya Virus Infection of *Aedes albopictus* Mosquitoes

**DOI:** 10.3390/v13040553

**Published:** 2021-03-25

**Authors:** Michele Marconcini, Elisa Pischedda, Vincent Houé, Umberto Palatini, Nabor Lozada-Chávez, Davide Sogliani, Anna-Bella Failloux, Mariangela Bonizzoni

**Affiliations:** 1Department of Biology and Biotechnology, University of Pavia, via Ferrata, 27100 Pavia, Italy; michele.marconcini01@universitadipavia.it (M.M.); elisa.pischedda01@universitadipavia.it (E.P.); umberto.palatini01@universitadipavia.it (U.P.); alejandro.chavez@unipv.it (N.L.-C.); davide.sogliani@gmail.com (D.S.); 2Arbovirus and Insect Vectors Unit, Department of Virology, Institut Pasteur, 25-28 Rue du Dr Roux, 75015 Paris, France; vincent.houe@pasteur.fr (V.H.); anna-bella.failloux@pasteur.fr (A.-B.F.)

**Keywords:** chikungunya virus, *Aedes albopictus*, RNAi, viral integration, vDNA

## Abstract

The Asian tiger mosquito *Aedes albopictus* is contributing to the (re)-emergence of Chikungunya virus (CHIKV). To gain insights into the molecular underpinning of viral persistence, which renders a mosquito a life-long vector, we coupled small RNA and whole genome sequencing approaches on carcasses and ovaries of mosquitoes sampled 14 days post CHIKV infection and investigated the profile of small RNAs and the presence of vDNA fragments. Since *Aedes* genomes harbor nonretroviral Endogenous Viral Elements (nrEVEs) which confers tolerance to cognate viral infections in ovaries, we also tested whether nrEVEs are formed after CHIKV infection. We show that while small interfering (si)RNAs are evenly distributed along the full viral genome, PIWI-interacting (pi)RNAs mostly arise from a ~1000 bp window, from which a unique vDNA fragment is identified. CHIKV infection does not result in the formation of new nrEVEs, but piRNAs derived from existing nrEVEs correlate with differential expression of an endogenous transcript. These results demonstrate that all three RNAi pathways contribute to the homeostasis during the late stage of CHIKV infection, but in different ways, ranging from directly targeting the viral sequence to regulating the expression of mosquito transcripts and expand the role of nrEVEs beyond immunity against cognate viruses.

## 1. Introduction

Chikungunya virus (CHIKV) was first isolated in Tanzania in 1952 [1] causing sporadic outbreaks. In 2005, it re-emerged in costal Kenya and spread to the islands of the Indian Ocean and India [2]. After 2005, CHIKV moved globally, causing outbreaks not only in tropical regions of the world such as South-East Asia, Central Africa, and Central America, but also in temperate Europe and North America [3,4,5]. The increasing public heath significance of CHIKV is tightly linked with the global invasion of Asian tiger mosquito *Aedes albopictus* which established in temperate areas of the world [6]. In response to this trend, chikungunya was included in the WHO list of the most relevant neglected tropical diseases in 2017 [7].

Adult female mosquitoes acquire viral pathogens through an infectious blood meal. Ingested viral particles must replicate in the midgut and further disseminate throughout the mosquito general cavity to reach salivary glands. Only if/when infection is established in the salivary glands, mosquitoes can transmit viruses to a new host during a subsequent blood meal, thus continuing the cycle [8,9]. In *Ae. albopictus*, CHIKV infection progresses rapidly and viral particles are detected in the saliva as early as 2 days post infection (dpi) [10]. During the early stages of infection, viral titer increases and mosquitoes activate a complex immunity response [11,12,13]. Rarely, viruses are blocked in the midgut (i.e., midgut infection barrier) or cannot disseminate from the midgut (midgut escape barrier) or the salivary glands (salivary glands barrier) [14,15]. These infection-refractory phenotypes are both viral species and mosquito strain/species specific [16,17,18,19]. Most frequently, a balance between viral replication and the mosquito immune system is reached, resulting in the establishment of a non-pathogenic persistent infection which ensures mosquitoes a life-long viral transmission capacity [20,21]. Thus, in the arboviral transmission cycle, there are at least two critical moments: the establishment of viral infection in the mosquito midgut and the shift from acute to persistent infection which renders a mosquito a life-long vector. The understanding of the molecular mechanisms underlying the ability of mosquitoes to acquire, maintain, and transmit arboviruses is expected to aid in development of novel genetic-based control strategies. With the aim of identifying strategies and/or effectors to control viral infection in mosquitoes, most studies focused on biological (i.e., expressional and/or metabolic) changes occurring in mosquitoes in the early time points post infection [22,23]. However, it was recently proposed that unravelling the molecular underpinning of viral persistence could facilitate the design of population replacement strategies of mosquito control which avoid emergence of viral resistance [24].

Both during the initial acute phase of infection and in persistent infection, the cornerstone of mosquito antiviral defense is RNA interference (RNAi) [25,26,27]. RNAi relies on small single-stranded RNA molecules of 20–30 nucleotides (nt) that recognize and act on target sequences based on sequence-complementarity and through interaction with Argonaute proteins [27,28]. Three classes of small RNAs are known, namely small interfering RNAs (siRNAs), microRNAs (miRNAs), and PIWI- interacting RNAs (piRNAs). These small RNAs differ in length, interacting proteins, mechanisms of biogenesis, targets, and functions ranging from direct target cleavage by Argonaute-2 guided by siRNAs to destabilization of target mRNAs or translation blockage for miRNAs or silencing of transposable elements at the transcriptional or translational levels for piRNAs [28]. Additionally, miRNA-target interaction may also lead to positive regulation of the target by increasing mRNA stability or action on promoter sequences [29]. Despite these differences, increasing experimental evidence reveals crosstalk among RNAi pathways, firstly between the siRNA and piRNA pathways, after arboviral infections of mosquitoes [30,31]. One of the links between the two pathways are viral DNA (vDNAs) fragments, which appear hours after arboviral infections and serve to amplify siRNA-mediated silencing [30,32,33]. In *Drosophila*, vDNA forms are produced by defective viral genomes through the helicase domain of Dicer-2, which also mediates siRNA biogenesis [33]. In *Ae. albopictus*, vDNA forms corresponding to the region of the CHIKV genome encompassing the nsP2 and nsP4 coding sequences have been detected up to 9 days post infection [30]. Moreover, in *Aedes* spp. mosquitoes, DNA fragments from insect-specific virus (ISVs), phylogenetically linked to arboviruses of the *Flaviviridae* and *Rhabdoviridae* families, are integrated into the genome [34,35,36]. These nonretroviral Endogenous Viral Elements (nrEVEs) are enriched in piRNA clusters, genomic regions from which piRNAs are generated, and produce piRNAs [34,35,36]. Selected nrEVEs of *Aedes aegypti* limit infection by cognate viruses through EVE-derived piRNAs [37,38], but the genome of neither *Ae. aegypti* nor *Ae. albopictus* has nrEVEs derived from CHIKV [35,36]. Additionally, the landscape of nrEVEs is variable not only across mosquitoes of different geographic populations, but also within laboratory-reared strains [39,40].

The analyses of the profile of small RNA molecules after arboviral infection not only showed differences among mosquito species and viral families [41], but also revealed that small RNA molecules may be produced from infecting viruses, including viral piRNAs (vpiRNAs) and, possibly, miRNAs deriving from pri-miRNAs stem loops which are processed directly in the cytoplasm [29,42,43,44,45]. Albeit still debated, arbovirus-encoded miRNA-like molecules appear to either regulate expression of mosquito genes (i.e., the transcription factor GATA4 by the West Nile virus-derived KUN-miR-1) or directly virus genes, leading to regulation of virus replication [46,47,48]. The role of vpiRNAs following arboviral infection has not been elucidated yet [45].

Besides changes in the abundance of small RNA molecules following arboviral infections, transcriptional changes occur following infections with CHIKV and dengue virus (DENV) [49]. 

Here, we combined the analysis of high throughput sequencing data (whole genome sequencing, WGS, and small RNA-seq) and molecular biology techniques to test for vDNA presence and new nrEVE formation and analyze the production of small RNAs, including those from nrEVEs existing in the *Ae. albopictus* genome, 14 days post CHIKV infection, when infection has become persistent [50]. We show that CHIKV infection does not result in new nrEVE formation, but the differential accumulation of piRNAs derived from existing nrEVEs correlates with differential expression of an endogenous transcript. The profile of siRNAs was even along the CHIKV genome. On the contrary, piRNAs mostly arise from a ~1000 bp window, which corresponds to the region encoding the C and E2 proteins. A unique vDNA fragment matching this region was also identified.

Statistically significant differentially accumulated siRNAs were also identified and correlated with differential abundance by qRT-PCR of selected, bioinformatically-deduced, target transcripts. Our results clearly show that RNAi pathways not only target the viral sequence, but host miRNAs and piRNAs are also modulated and contribute to cellular homeostasis during persistent CHIKV infection. The identified correlation between the differential accumulation of nrEVE-piRNAs and the decreased expression of the nrEVE-piRNA target expands the possible role of nrEVEs beyond immunity against cognate viruses.

## 2. Materials and Methods

### 2.1. Mosquito Infections

*Aedes albopictus* mosquitoes of the Foshan strain were used in this study. Mosquitoes were reared under constant conditions, at 28 °C and 70–80% relative humidity with a 12/12 h light/dark cycle. One-week old females were infected with CHIKV 06.21 as previously described [51]. CHIKV 06.21 was isolated in La Reunion island and was kindly provided by the French National Reference Center for Arboviruses. Thirty CHIKV-infected mosquitoes were collected 14 days post infection (dpi), along with un-infected controls. Ovaries were dissected from carcasses and the virus was titrated by fluorescence focus assay on C6/36 *Ae. albopictus* cells as previously described [51]. At 14 dpi, mean viral titer was 1515 ± 628 FFU/mL. Dissected ovaries and carcasses were pooled into groups of 15 giving a total of 8 samples, 2 pools for each condition (Appendix A). Each sample was used for DNA and RNA extraction using the Nucleospin tissue or the Nucleospin miRNA kit (Macherey-Nagel), respectively, following manufacturer’s instructions. A second infection experiment was performed as before, resulting in mean viral titer of 1589 ± 50 FFU/mL. Samples from this infection experiment were collected as before and were used for RNA extraction using the standard Trizol protocol.

### 2.2. Whole Genome and Small RNA Sequencing

After quality control, DNA of the first infection experiment was sent to the Beijing Genomics Institute (BGI) for Illumina library preparation. Libraries were run on the Illumina Novaseq 6000 by to obtain 150 bp paired end reads. Small RNA libraries were prepared by BGI from the same samples as for WGS and sequenced on the BGI-Seq50 to obtain 40 million reads per sample.

### 2.3. Detection of vDNA Fragments

The presence of vDNA fragments was investigated in CHIKV infected ovaries and carcasses by PCR using primers designed to cover the whole viral genome and producing overlapping amplicons of ~300–400 bp each (Appendix A). PCR-amplified bands were cloned using the TA cloning kit (Invitrogen) following manufacturer’s instruction and sequenced by Macrogen Europe (Amsterdam, The Netherlands).

### 2.4. Bioinformatic Analyses of WGS Data

For all genomic analyses regarding the *Ae. albopictus* genome, we used the latest genome assembly (Aalbo_primary.1, Refseq assembly GCF_006496715.1) [34]. WGS data provided by BGI were aligned to the genome of *Ae. albopictus* using BWA-MEM algorithm [52]. Because the landscape of nrEVEs in the genome of *Ae. albopictus* is not fixed, but mosquitoes may have different nrEVE patterns [40], we first assessed which nrEVEs were present in each sample using Genome Analysis ToolKit (GATK) [53] as previously described [40,53] and then assessed an unbiased small RNA profile of nrEVEs after CHIKV infection. Due to sequence similarity among nrEVEs, a nrEVE was considered absent in a sample when the average coverage for that nrEVE was less than 1 read with zero mapping quality. To identify possible viral integrations formed after CHIKV infection, we used the VyPer and ViR pipelines [54,55]. The similarity of nrEVEs sequences to CHIKV 06.21 was calculated by grouping nrEVEs depending on their viral family and then aligning them to the genome of CHIKV 06.21 (NC_004162.2) with MAFFT [56]. We retrieved the identity matrix as implemented by the EMBL-EBI search and sequence analysis tools [57] and plotted the results with custom scripts.

### 2.5. Bioinformatic Analyses of sRNA Data

For the following analyses, sRNA data were aligned using sRNAmapper [58] to either the genome of CHIKV 06.21 (NC_004162.2) or *Ae. albopictus* transcripts (Aalbo_primary.1, Refseq assembly GCF_006496715.1) using the “-best” option. Sequences between 20–22 nt were identified as siRNAs/miRNAs and sequences between 25–30 nt were identified as piRNAs. This classification of sRNAs was based on available literature [58,59,60,61,62], and through bioinformatics analysis as follows. First, we plotted the length distribution of all mature miRNAs reported for *Ae. aegypti*, downloaded from miRBase database v.22 [63], and corroborate that the major range for miRNAs occurs between 20 to 22 nt (Appendix A). Second, we discarded ranges of nts where miRNA and piRNA definitions overlap, as the tail distribution of miRNAs (size range of 23–24 nt) shows (Appendix A). Finally, taking into account that smaller peaks at 20 and 22 nt are often observed as Argonaute-2 products [64], and that thousands of 20 and 22 nt unique reads perfectly mapped to the viral genome (Figure 1), we decided that the broader range of 20–22 nt would provide less biased results than limiting the analyses to 21 nt reads (i.e., exo-siRNA consensus size).

#### 2.5.1. Small RNA Profile of the CHIKV Genome

sRNA data were mapped against the genome of CHIKV 06.21 (NC_004162.2). To visualize the coverage along the virus genome, alignment (bam) files of biological replicates were merged using bamCompare as implemented in DeepTools2 [59], normalizing the data to reads per kilobase of transcripts (RPKM) values based on total library size, setting bin size to 1 nt to improve resolution, and computing the mean of the number of reads between the files. Alignment results were exported as bedgraph files and visualized in Integrative Genomics Viewer (IGV) [60]. sRNAs were filtered by length using BBMap reformat.sh [65]. PingPongPro and the small RNA signature tool [61] were implemented in Galaxy [66] to check for ping-pong signature of reads identified as piRNAs; 1U and 10A biases were plotted with weblogo 2.8.2 [64]. siRNAs that had mapped to CHIKV (hereafter called viral-sRNA) were re-mapped to *Ae. albopictus* transcripts and visualized in IGV [60], after merging alignment (bam) files with samtools merge [67]. Candidate sequences of viral miRNA-like precursors were identified using ViralMir [68] and miRNAFold [69] with default parameters by analyzing the secondary structure of the 150 bp viral RNA sequence including candidate viral miRNA-like sequences.

#### 2.5.2. Small RNA Profile of Mosquito Transcripts, piRNA Clusters and nrEVEs

sRNA data were aligned to *Ae. albopictus* transcripts. Only sRNAs mapping in reverse orientation with respect to the predicted transcripts were retained. sRNAs were filtered by length using BBMap reformat.sh [65] as indicated above. Counts for each transcript were performed using the subread featureCounts.sh [70]. Transcripts displaying differential abundance of small RNAs between CHIKV infected and control samples were identified and checked against the list of *Ae. albopictus* immunity genes [34] to identify possible regulation of immunity pathways by sRNAs. sRNA data were aligned to the *Ae. albopictus* genome following the same criteria as above and their abundance in each piRNA-cluster was estimated using the subread featureCounts.sh [70]. Comparison between infected vs. non-infected samples in the abundance of sRNA mapping to transcripts (hereafter called T-sRNA) was established based on read counts in edgeR using the weighted trimmed mean of M-values with zero pairing (TMMwzp) normalization method and the likelihood ratio test to assess statistical significance [71]. As implemented in edgeR, genomic regions, or transcripts that had a minimum total read count of 15 across all samples were discarded [72]. Statistical significance was based on a 0.05 FDR threshold. Three normalization strategies were implemented to minimize bias in analysis of differential abundance. Normalizations were based on: (1) the total number of sRNA reads that mapped on the reference genome, (2) the total number of reads analyzed in edgeR, following size-filtering and counts, (3) the fraction of miRNA sized molecules that mapped onto a list of previously identified miRNAs [34], as performed in [73,74,75]. Results from the three normalization methods were compared and visualized in Venn diagrams using Venny 2.1.0 [76] showing that the normalization based on the total number of sRNA reads has fewer outliers than the other methods and fewer outliers (Appendix A). Transcripts displaying significant differential abundance of small RNAs between infected or uninfected were analyzed using Blast2GO [77]. Deduplicated target transcripts were blasted using default parameters against all currently available Culicidae genome sequences and the top 5 hits were retrieved. GO analyses were performed merging the results from the Interpro classification and the results from the blast database with the subsequent mapping and annotation, as implemented in Blast2GO. GO enrichment was performed based on the annotation of the full sets of transcripts of *Ae. albopictus* with FDR cut-off of 0.05. To find the most represented pathways, target transcripts were analyzed in KOBAS 3.0 [78], selecting *Ae. aegypti* as reference species and searching the Kyoto Encyclopedia of Genes and Genomes (KEGG) pathway (K option). Results were considered significant at *p*-value < 0.05.

Small RNA coverage of nrEVEs was tested after having verified nrEVE presence in the sample under study as described above. Despite the high stringency implemented in sRNAmapper (18 nt perfect seed match, 1 mismatch allowed after the seed and 2 mismatches allowed on the 3′ end tail of the read), single sRNA molecules could be mapped to several nrEVEs with equal confidence because of the high sequence similarity among nrEVEs [34]. To identify sRNAs mapping to nrEVEs and accumulate differentially across samples, we used two strategies to reduce any bias due to nrEVE sequence similarity: (1) we focused on single small RNAs (level 1) or (2) we took in consideration the overall small RNAs mapping to a nrEVE (level 2) (Appendix A). For the first strategy, reads from each condition that mapped to nrEVEs were collapsed based on their sequence identity with fastx-toolkit [79] and each sequence was used to count the amount of each piRNA and siRNA sequence present in every fastq file, using custom scripts. Individual sRNA names were given following fastx-toolkit results in the format “rank—number” where rank starts at 1 for sRNA with the highest read count and number is the actual number of reads. For the second strategy, piRNAs and siRNAs abundance from each nrEVE was estimated using subread featureCounts.sh [70]. Differential abundance of small RNAs mapping to nrEVEs (hereafter defined as EVE-sRNA) between infected vs. uninfected samples was assessed in edgeR as described in the previous paragraph. EVE-sRNAs that were found statistically differentially abundant between infected and not infected samples were used to blast (BLASTn, default settings) [80] the collection of *Ae. albopictus* transcripts [34]. When found, the top 10 blast hits of differentially-abundant EVE-sRNAs were retrieved and thermodynamic analyses on the RNA-RNA pairing were performed using the software IntaRNA locally with heuristic mode under the model B [81,82], setting the seed between positions 2–7 [83] and the seed energy threshold <−3.2 kcal/mol [81]. Candidate target transcripts were analyzed using Blast2GO [77] to find their Gene Ontology terms as previously described. Functional annotation was also performed in Argot 2.5 [84] searching against Culicidae with default parameters and a cut-off ≥200.

### 2.6. Differential Expression of Ae. albopictus Annotated miRNAs

sRNA data were mapped to the consensus sequences of 229 loci corresponding to 121 validated *Ae. albopictus* pre-miRNAs [34]. Counting and differential expression analysis were performed in edgeR as described above. Results of identical miRNAs from different loci were averaged. For each differentially expressed miRNA, both the consensus star sequence and the consensus mature sequence were retrieved, and the possible targets investigated using blast and IntaRNA as described for EVE-sRNAs. Reads that mapped on differentially expressed miRNAs were also aligned to the CHIKV genome to investigate their potential as antiviral miRNAs.

### 2.7. Quantitative PCR

Selected target transcripts from all previous analyses (i.e., from the list of differentially abundant T- sRNAs and transcripts targeted by differentially abundant EVE-sRNAs or viral-sRNAs) were tested for differential expression by quantitative PCR (qPCR) on samples from the second infection experiment. RNA from these samples was extracted using standard Trizol protocol and reverse transcribed with the GoScript reverse transcriptase kit (Promega), following manufacturer’s instructions. qRT-PCR reactions were performed using the QuantiNova SYBR Green PCR Kit (Qiagen) following the manufacturer’s instructions on an Eppendorf Mastercycler RealPlex4 (Appendix A). Estimates of relative quantification were performed with the delta-delta-Ct method implemented in the software qBase+ (Biogazelle), using RPL34 gene as housekeeping [85]. Statistical significance was assessed by ANOVA and unpaired t-tests as implemented in qBASE+. Selected piRNAs were also investigated using qPCR (Appendix A). Small RNA molecules from the same samples used for transcript quantification were reverse transcribed using the miScript II RT Kit (Qiagen) following manufacturer’s instruction with the HiSpec buffer to ensure only mature sRNAs would be selected. qPCR reactions were set up using the miScript SYBR Green PCR Kit (Qiagen), following manufacturer’s instruction. piRNA expression was normalized to the expression of U6B small nuclear RNA (RNU6B).

## 3. Results and Discussion

*Aedes albopictus* mosquitoes of the reference Foshan strain were infected with CHIKV. At 14 dpi, when persistence has established, saliva and two pools of each 15 female carcasses and ovaries were collected. Pools of carcasses and ovaries were processed to extract both DNA and RNA for WGS and small RNA-seq, respectively. WGS data were used to derive the pattern of nrEVEs of the sampled mosquitoes and verify whether CHIKV infection results in new nrEVEs formation. Small RNA-seq was used to quantify the changes in small RNAs against CHIKV, the *Ae. albopictus* transcriptome and nrEVEs (Appendix A).

WGS libraires resulted in a total of 1,324,733,838 and 1,713,328,214 clean reads in CHIKV-infected carcasses and ovaries, respectively, and 1,467,049,196 and 774,603,540 clean reads in uninfected carcasses and ovaries, respectively, after grouping biological replicates. Sequencing of small-RNA libraries resulted in a total of 1,551,205,753 and 1,010,736,055 reads, in the range of 18–30 nt, which could be mapped to the reference *Ae. albopictus* genome from infected ovaries and carcasses, respectively, after grouping biological replicates. Similarly, 1,568,888,349 and 1,651,306,629 reads were mapped for un-infected control ovaries and carcasses, respectively. Reads size distribution of ovaries samples shows a peak at 21 nt in both infected and uninfected samples, as well as marked piRNAs production peaking at 28 nt (Figure 1). In uninfected and CHIKV- infected carcasses samples, only a peak at 21 nt was seen, indicating a lower production of piRNAs in carcasses (Figure 1).

### 3.1. Small RNA Profile of CHIKV, vDNA Fragments, and New nrEVEs

The CHIKV genome consists of a single positive strand RNA of ~11–12 Kb. Upon entry into a host cell and uncoating, the CHIKV genome is used both as mRNA for translation of nonstructural proteins and as the template for the synthesis of a complementary negative strand RNA molecule, which is the replicative form of the virus. A genomic length positive strand molecule is synthesized from the negative strand RNA, along with a shorter RNA molecule from an internal promoter. This subgenomic positive strand RNA molecule encodes for nonstructural proteins [86]. Small RNA sequencing data from CHIKV-infected and control samples were mapped against the CHIKV genome. Mapped sRNAs were filtered by size to assess siRNA- (20–22 nt) and piRNA- (25–30 nt) responses. A total of 19,008 and 316,527 siRNA size reads mapped to the CHIKV genome from infected ovaries and carcasses, respectively (Figure 2A). Average strand-specific RPKM normalized counts show that small RNAs map predominantly on the positive strand of CHIKV, with the only exception of siRNAs in CHIKV-infected ovaries. This result indicates that the RNAi activity is mostly focused on the negative strand (Table 1).

Consistently, piRNA size reads were found to map against CHIKV (Figure 2B), with a predominance of piRNA size reads on the positive strand of the virus (Figure 2C). We found a total of 5510 piRNA size reads in ovaries and 60,824 in carcasses of CHIKV-infected samples with a clear pick between positions ~7600–8700 of the viral genome, corresponding to the region encoding the C and E2 proteins. A pick of piRNAs at the 5′ terminus of the CHIKV genome, around nucleotide position 8000, was previously identified in both *Ae. aegypti* and *Ae. albopictus* mosquitoes 4 dpi with CHIKV [87]. Accordingly, we saw a single vDNA form between positions 7964–8742 of the viral genome in infected ovaries (Figure 2B). A smaller fragment in the same region was found in carcasses. In a laboratory colony of *Ae. albopictus* derived from Vietnam, vDNA forms from nonstructural proteins (i.e., nsP2 and nsP4) were identified as early as 2 dpi and up to 9 dpi with CHIKV [30]. These vDNA forms were shown to influence the profile of siRNAs during the early phase of infection. Taken together, these results indicate that the subgenomic mRNA is a general hot spot for piRNAs in *Aedes* spp. mosquitoes both during acute and persistent CHIKV infection [42]. On the contrary, vDNA forms can be generated from different parts of the CHIKV genome, possibly depending on the combination of viral and mosquito strains. Additionally, while during the early phases of arboviral infection, vDNAs modulate siRNAs in mosquitoes [30,33], during persistent infection, we could identify a unique vDNA form corresponding to a pick of piRNAs. Overlap analysis of piRNA reads revealed that the ping-pong amplification loop is active at this stage of CHIKV infection both in ovaries and carcasses (Figure 2D).

Using WGS data from the same samples, the presence of novel nrEVEs was tested, implementing VyPER [54] followed by ViR [55], resulting in no signals. This result indicates that no integration of vDNA fragments occurred following CHIKV infection in either soma (carcass) or germline (ovary) cells. The absence of nrEVEs following CHIKV infection may be due to the fact that integrations of viral sequences is a continuous but rare event in *Aedes* genomes and may depend on the viral load and the frequency of infection [36]. Additionally, no nrEVE with similarity to CHIKV has been characterized in arthropod genomes so far [34,35,88], suggesting that vDNA forms of CHIKV may remain episomal. 

### 3.2. Viral SmallRNAs on Ae. albopictus Transcripts

After having identified small RNAs that mapped to CHIKV, we checked whether any of these viral small RNA molecules also mapped to *Ae. albopictus* transcripts. A total of 68 siRNA reads from the CHIKV-infected samples, deriving primarily from two hotspots on the nsP3 coding region (Figure 3A), mapped to *Ae. albopictus* transcripts with an enrichment for afadin-like protein (LOC115259774), glucose transporter type1-like proteins (LOC109428702—LOC109424279), and an uncharacterized protein (LOC115263499) (Appendix A). These siRNA reads and the corresponding CHIKV genomic sequences were further analyzed to check for hairpins using viralMir [68] and miRNAFold [69]. These small RNAs may originate from hairpin structures within the CHIKV genomic sequence (Appendix A). The expression levels of the above-mentioned three transcripts were compared by qPCR between CHIKV-infected samples and un-infected controls (Figure 3B). Results showed a significant decrease in expression for transcripts encoding the glucose transporter and the uncharacterized protein (*p*-values < 0.05), but not that of afadin-like protein, in infected carcasses. Overall, these results suggest that CHIKV encodes for small RNAs which may target mosquito transcripts. However, whether these small RNAs are canonical miRNAs or are part of the heterogenous population of siRNAs produced following infection has to be further investigated. Modulation of pathways linked to energy production, including carbohydrate metabolism, has already been observed in mosquitoes following arboviral infection [89,90,91]. It was proposed that induction of glucose metabolism following viral infection may compensate the lack of energy, which derive from the breakdown of mitochondrial membranes resulting in a lack of ATP [90,92].

### 3.3. The Profile of SmallRNAs on Ae. albopictus Transcripts

Modulation of mosquito small RNAs has been shown after arboviral infection not only as a direct response to the virus, but also as a way to maintain cellular homeostasis, by regulating gene expression [28,47,48]. On these bases, we mapped small RNA data to *Ae. albopictus* transcripts and kept only reads that mapped on antisense orientation to assess differential abundance between infected and uninfected samples. A total of 376,119 siRNA reads and 5,684,597 piRNA reads mapped to mosquito transcripts in CHIKV-infected ovaries; 1,084,531 siRNA reads and 690,008 piRNA reads in CHIKV-infected carcasses. Regarding the corresponding un-infected controls, 302,702 siRNA reads and 5,780,782 piRNA reads mapped to transcripts in ovaries, and 244,103 siRNA reads and 243,947 piRNA reads in carcasses. A total of 30 and 534 transcripts displayed significant differential abundance (SDA) of siRNAs in CHIKV-infected ovaries and carcasses, respectively, representing six over-represented GO terms, including “DNA helicase activity”, “DNA recombination”, and “DNA repair” (Appendix A). Interestingly, transcriptional regulation of genes associated with DNA repair and chromatin function has been recently linked to ovaries and carcasses, respectively [93] (Figure 4A,B, Appendix A).

GO enrichment analyses reveled an over representation representation of 6 terms, including “DNA helicase activity”, “DNA recombination”, and “DNA repair” (Appendix A). Interestingly, transcriptional regulation of genes associated with DNA repair and chromatin function has been recently linked to transgenerational immunopriming (TGIP) in *D. melanogaster* after infection with Sindbis virus, an Alphavirus-like CHIKV, which was clearly demonstrated only in females [93]. Thus, it is tempting to speculate that during persistent infection of female mosquitoes, there is transcriptional regulation of functions that not only helps with maintenance of infection, but also prepares to TGIP. Because piRNAs have been shown to contribute in regulating mosquito gene expression [94], we also checked for transcripts with significantly different piRNA profile between infected and uninfected samples. A total of 299 and 81 transcripts displayed SDA of piRNAs in CHIKV-infected ovaries and carcasses, respectively (Figure 4C,D, Appendix A). In carcasses, the majority of these transcripts showed lower abundance of piRNAs during infection. KOBAS gene-list enrichment analysis of these transcripts revealed an enrichment for several pathways, including oxidative phosphorylation, purine metabolism, and DNA damage-related pathways (Appendix A).

Two transcripts displayed differential abundance for piRNAs in CHIKV-infected ovaries and carcasses as well as for siRNAs in CHIKV-infected ovaries; four transcripts displayed differential abundance for piRNAs and siRNAs in CHIKV-infected ovaries; 23 transcripts displayed differential abundance for piRNAs and siRNAs in CHIKV-infected carcasses (Appendix A). The expression levels of 3 transcripts (XM_029853235.1, uncharacterized; XM_029877572.1 encoding for a neurofilament heavy polypeptide-like protein and XM_029878700.1 for a hydrolase), which had consistent differential abundance of small RNAs across conditions and the relatively highest number of reads, were compared by qPCR between infected and un-infected samples. qPCR results show all three transcripts are significantly under-expressed in carcasses during infection compared to un-infected controls (Figure 4E).

The hydrolase XM_029878700.1 was the only tested transcript with SDA of only piRNAs. These piRNAs were extracted and re-mapped to the *Ae. albopictus* genome after masking the genomic region encoding for XM_029878700.1, leading to the identification of genomic region NW_021838576.1:117401889-117402177. This region, which maps outside annotated piRNA clusters [34], shows high sequence homology to XM_029878700.1, but without annotated ORFs (Figure 4F). Annotation as “hydrolase” is a generic term as hydrolysis can be performed on different chemical bonds, including sugar glucosyl bonds and lipids. It is interesting to notice that sugar and lipid homeostasis is a common feature of *Aedes* spp. mosquitoes after infection not only with CHIKV, but also Flaviviruses [22,90,91,92,95,96,97], suggesting that the candidate transcript identified here should be further functionally investigated.

The database of *Ae. albopictus* miRNAs was recently published [34], so we took advantage of this resource to check for specific changes in mosquito miRNA expression levels during persistent CHIKV infection. A total of 14 miRNAs were detected as differentially modulated (Table 2), with the majority being upregulated (8) in ovaries but depleted in carcasses. All the DE miRNAs (except for miR-new6) have a homolog in *Ae. aegypti*. Differential expression levels ranged between ~0.20-fold for aal-miR-new6 and ~20-fold for aal-miR-210.

We compared these results to a previous report of differentially expressed miRNAs in *Ae. albopictus* cells [98] and found only one miRNA present in both analyses (aal-miR-1000). Interestingly, miR-1000 was reported to be under-expressed in cells 24 h post infection, whereas we found it over-expressed in adult mosquitoes 14 dpi. We searched the *Ae. albopictus* transcriptome for potential target sites of the DE miRNAs, as well as the corresponding star miRNAs, performing thermodynamic analyses on the RNA-RNA pairing of the top 10 blast hits for each miRNA. We found a total of 229 possible targets (140 from over- expressed miRNAs, 89 from under-expressed miRNAs) (Appendix A). KOBAS analysis revealed 12 significantly enriched pathways (Table 3), including the Toll and Imd pathways, suggesting the under regulation of these immunity pathways during persistent infection. “Lysine degradation” was also identified among the enriched pathways when looking at differentially abundant piRNAs (Appendix A).

We also tested for potential targets site of the DE miRNAs on the CHIKV genome, but no significant alignment was found, indicating that these miRNAs cannot directly target CHIKV. Overall, these results show a different modulation of miRNAs in ovaries and carcasses. Depletion of miRNAs in carcasses is a trend consistent with what has been observed following Flavivirus infection (i.e., Zika and DENV) [99,100].

### 3.4. Differential Abundance of nrEVEs-sRNAs Following Infection and Their Targets on the Ae. albopictus Transcriptome

The genome of *Ae. albopictus* harbors hundreds of nrEVEs, which are variably distributed among the genomes of Foshan mosquitoes [40]. None of these nrEVEs derive from Alphaviruses and the maximum sequence similarity between annotated nrEVE and CHIKV is 70% (Figure 5A). Nevertheless, nrEVEs produce piRNAs [35] and responder and trailer piRNAs are produced from viral RNAs guided by endogenous piRNAs, indicating that piRNA biogenesis can spread outside a putative nrEVE targeted region [31,101]. On this basis, we first verified which nrEVEs our mosquitoes had, then tested whether the piRNA profile of these nrEVEs was different in infected vs. un-infected-fed samples and finally checked for putative targets of the differentially accumulated nrEVEs-derived piRNAs.

Differences in the piRNA profile of nrEVEs were observed only in CHIKV-infected ovaries. A total of 24 nrEVEs mapping in different clusters showed a piRNA profile that was significantly different between infected and un-infected samples (Appendix A). piRNAs from these nrEVEs peaked between 27 and 29 nt (Figure 5C). When looking at single piRNAs (Appendix A, level 1), a total of 34 and 4 piRNA molecules were significantly differentially expressed in infected vs un-infected ovaries and carcasses (Figure 5B), respectively, and peaked at 27 nt (Figure 5C). Of these 38 piRNAs, 30 originated from nrEVEs embedded in piRNA clusters. We focused on CHIKV-infected ovaries. Intersecting the genomic origins of the DE piRNAs with the SDA nrEVEs, we identified 24 DE piRNAs from 8 SDA nrEVEs (Appendix A). These 24 piRNAs do not blast on the CHIKV genome indicating that they do not target the viral mRNA, but they blast against *Ae. albopictus* transcripts. The free energy of the seed pairing between these piRNAs and their candidate target transcripts was evaluated with IntaRNA, showing that these piRNAs could interfere with the detected transcripts. Two piRNAs (1049-820 and 3135-266), both originating from nrEVE Rhabdo22 (Figure 5D, red bar), which is embedded in piRNA cluster 461, have a common target, transcript XM_029880513.1, an uncharacterized protein (Figure 5E). In accordance with bioinformatic data, qPCR analysis confirmed a higher expression of 3135-266 in infected vs. un-infected ovaries and a lower expression of the candidate target transcript (Figure 5F). Taken together, these results suggest that nrEVEs-derived piRNAs can be differentially accumulated during CHIKV infection and regulate the expression profile of endogenous transcripts.

## 4. Conclusions

Here, we describe the profile of small RNAs and the presence of vDNA forms and nrEVEs during persistent infection of *Ae. albopictus* with CHIKV. Because RNAi is the main antiviral pathway of *Aedes* spp. mosquitoes, a number of studies have described the profile of small RNAs following infection with arboviruses, including CHIKV [87,98,99,100,102,103,104,105]. In line with previous reports, we found a strong exogenous siRNA response throughout the whole CHIKV genome in both ovaries and carcasses [30,88]. On the contrary piRNAs aligned primarily to a region encompassing the C and E2 coding sequences from which a unique vDNA form was also identified, in both analyzed tissues. This result differs from previous studies in which various vDNAs from multiple regions of the genome of the flaviviruses Zika, dengue, and West Nile virus were identified 24 h post infection and up to 21 dpi in mosquito cells and adults, when infection has become persistent [30,32,106]. Whether this discrepancy is dependent on the specific viral species and mosquito species/strains studied or it indicates that only the vDNA molecule producing the most piRNAs is retained once infection has become persistent requires further investigations.

CHIKV infection did not result in the formation of novel viral integrations in the genome of *Ae. albopictus*. However, the piRNA profile of nrEVEs in the genomes of the tested mosquitoes changed and two nrEVE-derived piRNAs were further validated by qRT-PCR to be more abundant in CHIKV-infected than uninfected ovaries. Accordingly, the transcript to which these piRNAs map showed decreased in expression in CHIKV-infected ovaries. Thus, nrEVEs may have other functions than antagonistic action against cognate viruses, which was demonstrated in ovaries [38]. This result may explain previous findings describing that the presence/absence of certain nrEVEs correlates with differences in dissemination efficiency for CHIKV and dengue viruses, even if the genome of neither *Ae. aegypti* not *Ae. albopictus* has nrEVEs derived from these arboviruses [34,35,39]. To conclude, here we provided an in-depth analysis of small RNAs in CHIKV-infected *Ae. albopictus* mosquitoes at 14 dpi, when CHIKV has established persistent infection and identify target transcripts of differentially abundant small RNAs that merits further investigations. These results clearly show that the activity of the RNAi pathway modulates sRNAs and piRNAs, which contribute to cellular homeostasis, affecting primarily functions related to lipid and sugar metabolisms and DNA repair/binding. Finally, the identified correlation between the differential accumulation nrEVE-piRNAs and the decreased expression of the piRNA-target expands the possible role of nrEVEs beyond immunity against cognate viruses.

## Figures and Tables

**Figure 1 viruses-13-00553-f001:**
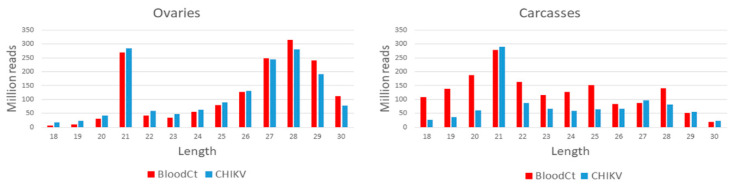
Size distribution of small RNAs reads mapping to the *Aedes albopictus* genome from ovary and carcass samples. Reads from chikungunya (CHIKV)-infected and uninfected samples are in blue and red, respectively.

**Figure 2 viruses-13-00553-f002:**
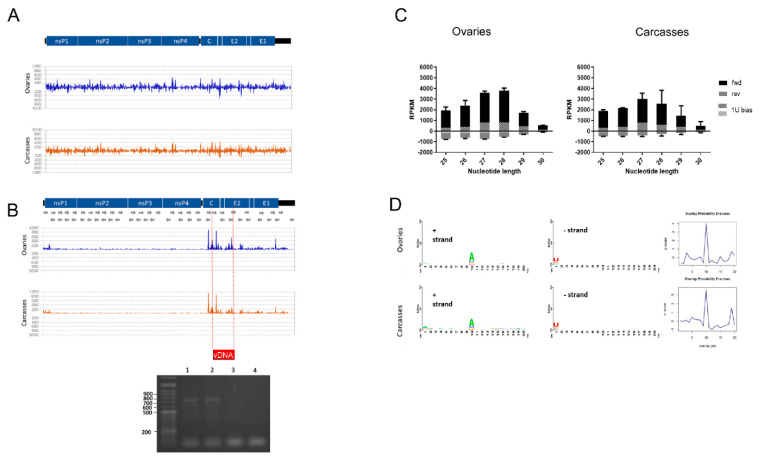
Small RNA response against the infecting CHIKV. (**A**) siRNA profile along the CHIKV genome. Values on the y-axis represent quantification of the signal as Reads Per Kilobase Million (RPKM). (**B**) piRNA profile along the CHIKV genome and PCR amplification of the vDNA fragment. Wells 1 and 2 show the PCR results from the DNA of the two biological replicates of CHIKV-infected ovaries; wells 3–4 show the PCR results from the DNA of the corresponding un-infected controls. The red box indicates the position in which the vDNA fragment was identified. (**C**) Read size distribution of piRNA reads in forward (black), reverse (grey), and displaying 1U bias (dots). (**D**) Nucleotide bias at each position of piRNAs against CHIKV mapping to the sense (+) strand (left panel) and antisense (−) strand (right panel) in ovaries and carcasses, and corresponding PingPong overlap probabilities.

**Figure 3 viruses-13-00553-f003:**
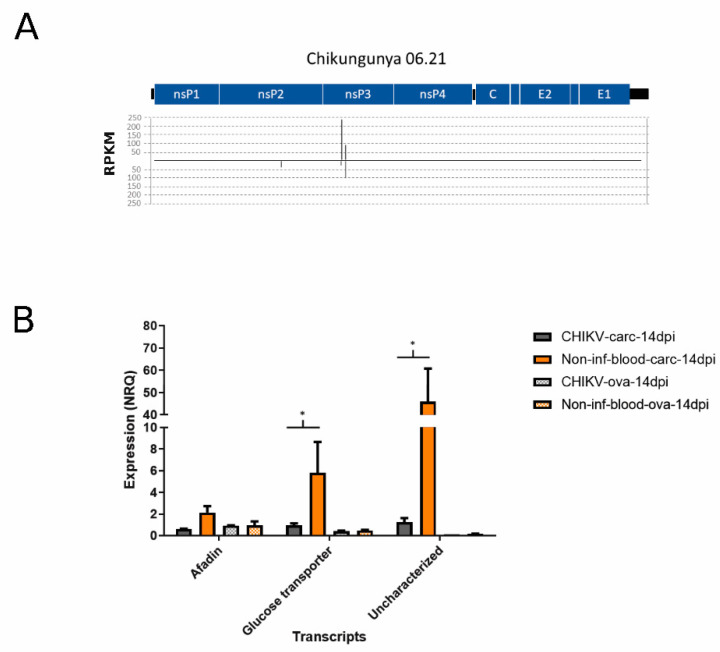
(**A**) Graphical representation of the hotspots on the viral genome for small RNA molecules that map to both CHIKV and *Ae. albopictus* transcripts. (**B**) Bar plot of the qPCR results on three candidate target transcripts of viral small RNA molecules: afadin (XM_029860564.1), glucose- transporter (XM_029879742.1), and uncharacterized transcript (XM_029866737.1). ANOVA significant results at *p* < 0.05 are indicated by an asterisk.

**Figure 4 viruses-13-00553-f004:**
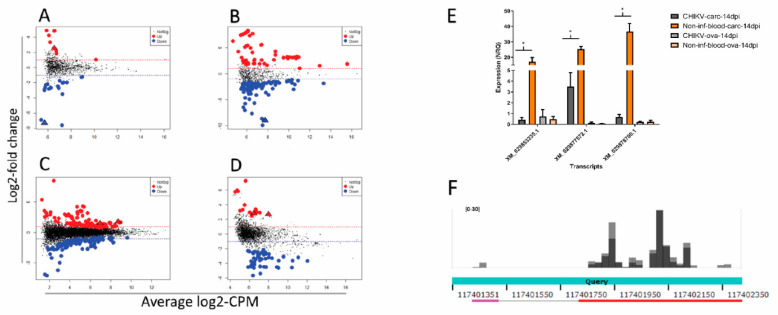
Differential abundance of small RNA molecules mapping to *Ae. albopictus* transcripts represented by average log counts-per-million (CPM) on the x-axis and log fold-change on the y- axis. Red dots indicate significantly over-expressed sRNAs and blue dots significantly under-expressed sRNAs upon infection compared to un-infected controls. Statistical significance of the data was assessed by likelihood ratio test. (**A**) siRNA abundance of transcripts in CHIKV-infected ovaries; (**B**) siRNA abundance of transcripts in CHIKV-infected carcasses; (**C**) piRNA abundance of transcripts in CHIKV-infected ovaries; (**D**) piRNA abundance of transcripts in CHIKV-infected carcasses. Triangles indicate transcripts tested by qPCR. (**E**) Bar plot of the qPCR results on transcripts to which SDA small RNAs are mapped to. Significant results at *p* < 0.05 are indicated by an asterisk. Significance was established by ANOVA. (**F**) Normalized coverage of the *Ae. albopictus* genomic region NW_021838576.1:117401889-117402177. The two shades of grey represent the biological replicates. Query bar indicates alignment similarity to XM_029878700.1 as computed by BLASTn online.

**Figure 5 viruses-13-00553-f005:**
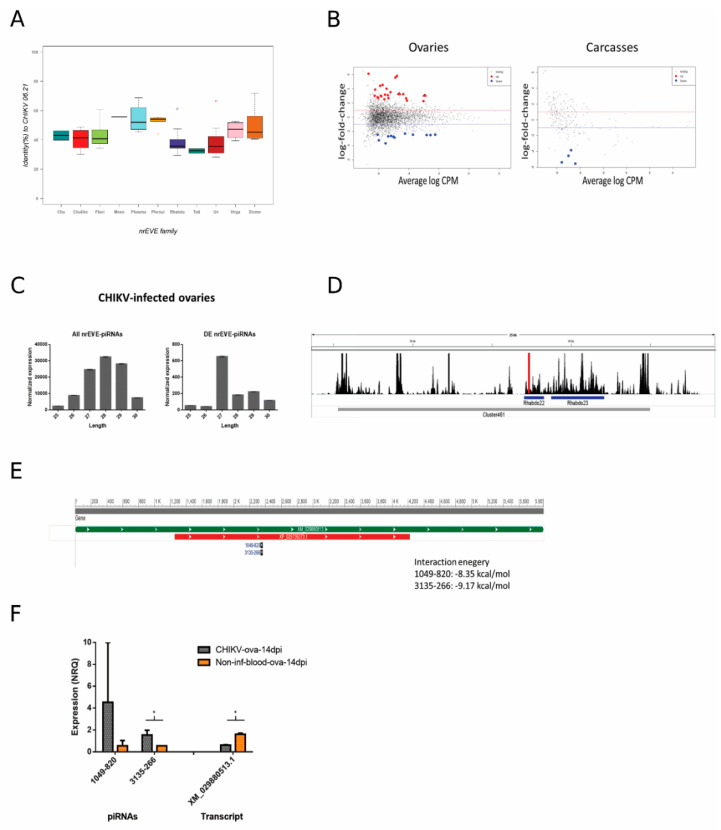
(**A**) Box plot representing nonretroviral Endogenous Viral Element (nrEVE) sequence identity with respect to the CHIKV genome. Chu stands for *Chuviridae* (dark blue); Chulike stands for *Chuviridae*-like (red); Flavi stands for *Flaviviridae* (light green); Meso stands for *Mesoniviridae* (gray); Phasma stands for *Phasmaviridae* (light blue); Phenui stands for *Phenuiviridae* (light orange); Rhabdo stands for *Rhabdoviridae* (purple); Toti stands for *Totiviridae* (dark green); Un stands for unclassified (dark red); Virgo stands for *Virgoviridae* (pink); Xinmo stands for *Xinmoviridae* (dark orange). (**B**) Differential abundance of piRNAs from nrEVEs in CHIKV-infected samples represented by average log counts-per-million (CPM) on the x-axis and log fold-change on the y-axis. Red dots indicate significantly over-expressed piRNAs and blue dots significantly under-expressed piRNAs upon infection compared to un-infected controls. Significance was assessed by likelihood ratio test. Triangles indicate piRNAs tested by qPCR. (**C**) Size distribution of piRNAs in CHIKV-infected ovaries. Left: size distribution of all piRNAs derived from nrEVEs. Right: size distribution of differentially abundant piRNAs derived from nrEVEs. Values are normalized according to the TMMwzp normalization method implemented in edgeR. (**D**) piRNA expression levels during infection in cluster visualized in IGV; nrEVEs in this region are highlighted (Rhabdo22 and Rhabdo23) as well as the peak representing the piRNAs used for qPCR analysis. (**E**) Graphical sequence panel (NCBI) representation of transcript XM_029880513.1; the piRNA target sites are represented on the bottom along with interaction energy as computed by IntaRNA. (**F**) Bar plot of the qPCR results on selected piRNAs and their putative target transcript. Significant results at *p* < 0.05 are indicated by an asterisk.

**Table 1 viruses-13-00553-t001:** RPKM values of strand-specific piRNA and siRNA reads which mapped to CHIKV.

	piRNAs		siRNA	
Sample	+RPKM	−RPKM	+RPKM	−RPKM
Ovaries	14,025 ± 1007	3036 ± 61	28,453 ± 1308	33,094 ± 766
Carcasses	11,610 ± 2993	1852 ± 613	38,731 ± 2230	26,837 ± 127

**Table 2 viruses-13-00553-t002:** Differentially expressed miRNAs in ovaries and carcasses. Total counts are calculated across sequencing libraries of infected replicates. Expression values are expressed as log2 transformed fold-changes (infected/control).

miRNA	Total Count	Log2FC	FDR
**Ovaries**			
aal-miR-210	791	4.396	0.003
aal-miR-124	2592	4.209	0.001
aal-miR-1000	1147	3.177	0.014
aal-miR-219	115	2.980	0.014
aal-miR-932	2858	2.890	0.001
aal-miR-981b	260	2.864	0.026
aal-miR-193	93	2.725	0.028
aal-miR-285	2257	2.179	0.012
aal-miR-2941	94,365	−1.026	0.026
aal-miR-7	6849	−1.054	0.012
aal-miR-316	2558	−1.200	0.012
aal-miR-9b	4903	−1.247	0.006
**Carcasses**			
aal-miR-new6	235	−2.399	0.016
aal-miR-1891	15,269	−1.929	0.016

**Table 3 viruses-13-00553-t003:** Biological pathways affected by differentially accumulated miRNAs. Biological pathways based on the Kyoto Encyclopedia of Genes and Genomes (KEGG) database of transcripts identified as targets of differentially abundant miRNAs. The list of target transcripts was obtained by KEGG Orthology Based Annotation System (KOBAS) gene-list enrichment analysis.

Term	Total Count	KEGG ID	*p*-Value
**Over-Expressed miRNAs Targets**			
Lysine degradation	9	aag00310	0.000
mTOR signaling pathway	9	aag04150	0.000
Toll and Imd signaling pathway	3	aag04624	0.003
Phosphatidylinositol signaling system	3	aag04070	0.006
Apoptosis	3	aag04214	0.007
Terpenoid backbone biosynthesis	2	aag00900	0.014
AGE-RAGE signaling pathway	2	aag04933	0.019
MAPK signaling pathway	3	aag04013	0.022
Endocytosis	3	aag04144	0.045
**Under-Expressed miRNAs Targets**			
SNARE interactions in vesicular transport	3	aag04130	0.000
Proteasome	3	aag03050	0.001
Phagosome	3	aag04145	0.003

## Data Availability

WGS data were deposited in the Sequence Read Archive (SRA) under Bioproject: PRJNA659388. Small RNA data were deposited in SRA under Bioproject: PRJNA607026.

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
