# Peer review of "Profile of Small RNAs, vDNA Forms and Viral Integrations in Late Chikungunya Virus Infection of Aedes albopictus Mosquitoes"

_viruses, 2021, doi:10.3390/v13040553_

Round 1
Reviewer 1 Report
This manuscript describes the profile of small RNAs, vDNA forms and nrEVEs 14 days post infection of Ae.albopictus with CHIKV. Overall the paper is well written.
Specific points to address are below:
- Line 132, it’s unclear what the study design is for mosquito infections. The authors note ovaries and carcasses gave a total of 12 samples but it’s unclear how 12 samples are derived from 30 mosquitoes. And also, the number of study time points are not clearly specified in the study design (figure S1A). The authors note there were “2 pools for each time point”, from what I can get there is only one time point, i.e., 14 dpi.
- Please check- Figure 2C, 1U bias has almost indistinguishable color/pattern from fwd/rev. Figure 3B and Figure 4E, consider changing the color/pattern for either ovaries or carcasses to make them more distinguishable.
- Table 2 and Table 3, numbers are reported to excessive precision (6 decimal places). Since the authors are just using 0.05 as a cut-off value, too many digits are not providing more information but can swamp the reader.
- Some editorial changes have been made in the annotated PDF.

Author Response
We would like to thank the reviewer for the constructive suggestions.
Point-to-point answers to the comments are below. Our answers are in italicus.
- Line 132, it’s unclear what the study design is for mosquito infections. The authors note ovaries and carcasses gave a total of 12 samples but it’s unclear how 12 samples are derived from 30 mosquitoes. And also, the number of study time points are not clearly specified in the study design (figure S1A). The authors note there were “2 pools for each time point”, from what I can get there is only one time point, i.e., 14 dpi.
REPLY: Thank you for noticing this incongruence. We sampled carcasses and ovaries 14 dpi, from CHIKV-infected and blood-uninfected samples. For each condition we had 2 replicates each consisting of 15 mosquitoes, resulting in 8 samples. Text was modified.
- Please check- Figure 2C, 1U bias has almost indistinguishable color/pattern from fwd/rev. Figure 3B and Figure 4E, consider changing the color/pattern for either ovaries or carcasses to make them more distinguishable.
REPLY: Color/pattern in Fig. 2C, 3B and 4E was modified to make items more distinguishable.
- Table 2 and Table 3, numbers are reported to excessive precision (6 decimal places). Since the authors are just using 0.05 as a cut-off value, too many digits are not providing more information but can swamp the reader.
REPLY: Values in Table 2 and Table 3 were changed.
- Some editorial changes have been made in the annotated PDF.
REPLY: Thank you very much for all the suggestions. We welcomed them all.
Reviewer 2 Report
The data presented in this paper are new and interesting. They may contribute to the understanding of mechanisms that participate on the regulation of CHIKV persistence and permissivity in mosquitoes and may have impact on the regulation of the virus transmission. Methodical approach is adequate to the aim of the study and is described in detail. The paper presents a lot of results that are documented by a lot of illustrative figures and tables. I would appreciate dividing of the results and discussion into separate sections would contribute to lucidity of the paper and to better significance evaluation of the results obtained.
Minor comments:
I recommend to use “ un-infected controls” instead of “blood -fed controls”
M+M: Why different RNA extraction protocol had been used in repeated infection experiment? For which purpose Trizol-isolated RNA was used?
Line 372: ….those coding for afadin-like protein….
Line 385: ….has already been observed in mosquitoes…
Line 405-407: what does it mean “CHIKV- infected representation of 6 terms” ?
Some abbreviations are not sufficiently explained (i.e. “T-sRNAs” , “dpi”, “DE miRNAs”)
Fig.5A lacks explanation of the symbols used
Author Response
We would like to thank the reviewer for the useful and constructive comments which allowed us to significantly improve our ms. Our point-to-point are below (in italicus).
- The data presented in this paper are new and interesting. They may contribute to the understanding of mechanisms that participate on the regulation of CHIKV persistence and permissivity in mosquitoes and may have impact on the regulation of the virus transmission. Methodical approach is adequate to the aim of the study and is described in detail. The paper presents a lot of results that are documented by a lot of illustrative figures and tables. I would appreciate dividing of the results and discussion into separate sections would contribute to lucidity of the paper and to better significance evaluation of the results obtained.
REPLY. Thank you for your comments. Because we have different levels of analyses (i.e. vDNAs, nrEVEs, smallRNA profile) we felt it was clearer to discuss each result section separately. We have a conclusion section that summarises and discuss again the main results.
Minor comments:
- I recommend to use “ un-infected controls” instead of “blood -fed controls”
REPLY: Thank you for your suggestion. Blood-fed controls was changed to un-infected controls.
- M+M: Why different RNA extraction protocol had been used in repeated infection experiment? For which purpose Trizol-isolated RNA was used?
REPLY: Trizol-isolated RNA was used for qPCR. The Nucleospin miRNA kit was used for RNA for smallRNA-sequencing libraries.
Line 372: ….those coding for afadin-like protein….
Line 385: ….has already been observed in mosquitoes…
Line 405-407: what does it mean “CHIKV- infected representation of 6 terms” ?
Some abbreviations are not sufficiently explained (i.e. “T-sRNAs” , “dpi”, “DE miRNAs”)
Fig.5A lacks explanation of the symbols used
REPLY: Thank you for these suggestions, we modified the text accordingly. Abbreviations were explained the first time they were cited.
Reviewer 3 Report
This paper of Marconcini et al. is very intresting since it deals with an emerging topic as the interplay between viral infections and sRNA expression with an elegant approach. Of course, as stted by the authors, there maybe some future work to perform with molecular techniques, but the bioinformatic results seem to be well promising.
I think the paper is quite well written, and my main perplexity is upon the choice to present many of the results inside the methods chapter, and the remaining results fused together with the discussion. This probably doesn' add to the readability and undertandability of the paper. Furthermore, since such a paper will be of interest also for people with no particular exprerience in bioinformatics, it would be better to simplify some parts of the methods explanation with this in mind (if possible. Maybe a graphic workflow overview may help).
Other suggestions: please remember to add information (reference) about all software used in the study, and try not to add excess references to single papers (i.e. ref 34).
Some minor points:
- line 5: Please check for the spelling of the name of co-author NLC.
- line 46: 'seldom'
- line 115: 'statistically significant, differentially accumulated'
- line 125: ref [40] is not relevant at this point of the text
- line 127: though the reference is correctly indicated, some indications about the viral strain should be present also in the text.
- line 130: 'fluorescence focus assay'
- Fig. S1A: in the control group the number is indicated as 30+30.Is this correct?
- line 136: 'as before'
- line 139: there is not enough indications on the methods of WGS, please add some information or references.
- line 150: add some reference for Macrogen Inc. (city,nation).
- line 154: 'BWA-MEM algorithm'
- line 157: GATK is a software, please add reference about the developers. The same for all other software tools used in the study and referred to from this point on.
- line 173: 'as follows'
- line 177: Fig. S2 doesn't show piRNAs size distriubution, only miRNAs.
- line 186: RPKM: please explain the acronym when first used.
- line 189: references 65 and 69 are the same. In any case, ref 69 would be in the wrong position in this part of the text.
- line 204: the presence in the text of some more information on how the list of immunity genes was obtained would be appreciated
- line 218: Ref. 34 doesn't seem to show any molecular miRNA validation, but only in silico validation of putative miRNAs. So, 'a list of previously identified miRNAs' would be a more appropriate sentence.
- line 218: 'as performed in'
- line 220: normalization method 1 has only fewer outliers, not fewer differences (differences are reciprocal).
- line 222: 'infected or uninfected samples'
- line 242: Ref. 79 is written in an unconventional form, since it has authors, as a publication, and an url as a tool. Please choose one form. Furthermore, the url is duplicated.
- line 242: 'each sequence was used...'
- line 268: could the authors please add the list of primers used for qPCR (at least in the supplementary files?).
- line 285: ref. [86] is not particularly relevant for this topic. RBU6B is used a normalizator for miRNAs, not piRNAs (and in the case of miRNA, there may be many other possible references to choose from).
- line 329: '5' terminus'
- line 372: Have also LOC109414992 and LOC109415135( 2nd ranked in RPKM) been taken into consideration?
- line 378: Fig.3B is somehow unclear: which condition has been used as reference? Is it explainable why these genes are not expressed in ovaries in any conditions? Furthermore, the 4-conditions/2-colors outfit of the graphic is not the best option. I complexively suggest to modify the graph in a more understandable figure.
- line 385: 'has already been observed'
- line 406: there is some errors in phrasing, probably part of the period has been switched with part of the following one.Please check.
- line 414: ' D. melanogaster'
- line 421: 'Fig. 4C-D'
- line 422: Ref [69] is not relevant in this period.
- line 450: Same as for Fig 3B: a part from the graphic look, it is not clear why gene expression is absent in ovary, and which of the conditions is the calibrator (in DDCT it must have a NRQ of 1)
- line 524: 'infected ovaries vs uninfected samples'
- line 526 '(Table S9)'
- line 527-528: 'in infected vs against CHiKV': not clear, re-write.
- line 536: Fig 5F: same as for the previous ones about the unclear choice of calibrators.
- line 553: 'the most piRNA' . There is some missing words, please add.
Author Response
We would like to thank the reviewer for having provided constructive suggestions. We thoroughly modified the ms according to his/her comments. Our point-to-point answers tare below, in italicus.
This paper of Marconcini et al. is very intresting since it deals with an emerging topic as the interplay between viral infections and sRNA expression with an elegant approach. Of course, as stted by the authors, there maybe some future work to perform with molecular techniques, but the bioinformatic results seem to be well promising.
I think the paper is quite well written, and my main perplexity is upon the choice to present many of the results inside the methods chapter, and the remaining results fused together with the discussion. This probably doesn' add to the readability and undertandability of the paper. Furthermore, since such a paper will be of interest also for people with no particular exprerience in bioinformatics, it would be better to simplify some parts of the methods explanation with this in mind (if possible. Maybe a graphic workflow overview may help).
Other suggestions: please remember to add information (reference) about all software used in the study, and try not to add excess references to single papers (i.e. ref 34).
REPLY: Because we have different levels of analyses (vDNAs, SmallRNA profile against CHIKV, Ae. albopictus transcripts and EVEs and new nrEVEs), we reasoned that discussing each result separately would allow readers to understand better the context and appreciate our results. We have a conclusion section where we highlight our main results and further discuss them.
Some minor points:
- line 5: Please check for the spelling of the name of co-author NLC.
REPLY: corrected
- line 46: 'seldom'
REPLY: changed to “rarely”
- line 115: 'statistically significant, differentially accumulated'
REPLY: corrected
- line 125: ref [40] is not relevant at this point of the text
REPLY: eliminated
- line 127: though the reference is correctly indicated, some indications about the viral strain should be present also in the text.
REPLY: details on the viral strain were included
- line 130: 'fluorescence focus assay'
REPLY: corrected
- Fig. S1A: in the control group the number is indicated as 30+30.Is this correct?
REPLY: Reviewer is correct
- line 136: 'as before'
REPLY: corrected
- line 139: there is not enough indications on the methods of WGS, please add some information or references.
REPLY: we rephrased the sentence to make it cleared. Illumina library preparation and sequencing on Novaseq 6000 is a standard procedure.
- line 150: add some reference for Macrogen Inc. (city,nation)
REPLY: corrected
- line 154: 'BWA-MEM algorithm'
REPLY: corrected
- line 157: GATK is a software, please add reference about the developers. The same for all other software tools used in the study and referred to from this point on.
REPLY: corrected
- line 173: 'as follows'
REPLY: corrected
- line 177: Fig. S2 doesn't show piRNAs size distriubution, only miRNAs.
REPLY: Reviewer is correct. We referred to Fig. S2 simply to highlight that miRNAs in the size range of piRNAs (23-24 nt) exist. The sentence was re-phrased to make this clearer.
- line 186: RPKM: please explain the acronym when first used
REPLY: corrected
- line 189: references 65 and 69 are the same. In any case, ref 69 would be in the wrong position in this part of the text.
REPLY: Thank you for noticing this error. The correct reference for PingPongPro was included.
- line 204: the presence in the text of some more information on how the list of immunity genes was obtained would be appreciated
REPLY: The list of immunity gene was as described in the Ae. albopictus genome paper that is included as reference, Ref 34.
- line 218: Ref. 34 doesn't seem to show any molecular miRNA validation, but only in silico validation of putative miRNAs. So, 'a list of previously identified miRNAs' would be a more appropriate sentence.
REPLY: Corrected
- line 218: 'as performed in'
REPLY: corrected
- line 220: normalization method 1 has only fewer outliers, not fewer differences (differences are reciprocal).
REPLY: Thank you. Corrected
- line 222: 'infected or uninfected samples'
REPLY: corrected, also following suggestions from Rev2.
- line 242: Ref. 79 is written in an unconventional form, since it has authors, as a publication, and an url as a tool. Please choose one form. Furthermore, the url is duplicated.
REPLY: corrected
- line 242: 'each sequence was used...'
REPLY: corrected
- line 268: could the authors please add the list of primers used for qPCR (at least in the supplementary files?).
REPLY: primers used for qPCR analyses were included in Table S1
- line 285: ref. [86] is not particularly relevant for this topic. RBU6B is used a normalizator for miRNAs, not piRNAs (and in the case of miRNA, there may be many other possible references to choose from).
REPLY: Ref86 was eliminated
- line 329: '5' terminus'
REPLY: Corrected
- line 372: Have also LOC109414992 and LOC109415135( 2nd ranked in RPKM) been taken into consideration?
REPLY: In this work, we focused the three transcripts listed in the text.
- line 378: Fig.3B is somehow unclear: which condition has been used as reference? Is it explainable why these genes are not expressed in ovaries in any conditions? Furthermore, the 4-conditions/2-colors outfit of the graphic is not the best option. I complexify suggest to modify the graph in a more understandable figure.
REPLY: The color/pattern of Fig.3B has been changed following also recommendations from Rev. 1. Expression analyses used the RPL34 as a reference as detailed in the method section.
- line 385: 'has already been observed'
REPLY: Corrected
- line 406: there is some errors in phrasing, probably part of the period has been switched with part of the following one.Please check.
REPLY: Corrected
- line 414: ' D. melanogaster'
REPLY: Corrected
- line 421: 'Fig. 4C-D'
REPLY: Corrected
- line 422: Ref [69] is not relevant in this period.
REPLY: Delated
- line 450: Same as for Fig 3B: a part from the graphic look, it is not clear why gene expression is absent in ovary, and which of the conditions is the calibrator (in DDCT it must have a NRQ of 1)
REPLY: The color/pattern of Fig.4E has been changed following also recommendations from Rev. 1. Expression analyses used the RPL34 as a reference as detailed in the method section. Differences between ovaries and carcasses are intriguing but require further experiments to understand the causes.
- line 524: 'infected ovaries vs uninfected samples'
REPLY: Delated
- line 526 '(Table S9)'
REPLY: Delated
- line 527-528: 'in infected vs against CHiKV': not clear, re-write.
REPLY: Re-phrased
- line 536: Fig 5F: same as for the previous ones about the unclear choice of calibrators.
REPLY: The color/pattern of Fig.5E has been changed following also recommendations from Rev. 1. Expression analyses used the RPL34 as a reference as detailed in the method section.
- line 553: 'the most piRNA' . There is some missing words, please add.
REPLY: Corrected as part of the sentence that was re-phrased (see comments on lines 527-528)